# HINTs: Human-INTuited Cues for Reinforcement Learning

## Abstract

In real-world scenarios, robots can leverage embodied reinforcement learning (RL) agents to solve continuous control problems that are difficult to model under partial observability. Especially when the control inputs are high-dimensional, RL agents can require extensive experience to learn correct mappings from the input space to action space, a serious limitation given the expense of developing sufficiently large real-world robotics datasets. Recent work approaches this problem by training agents in synthetic data domains or bootstrapping learning with direct human supervision. They are often difficult to apply to the target domain due to large distribution shift between the training and deployment setting (Zhao et al., 2020; Chen et al., 2022; Chae et al., 2022). We propose a novel learning framework, called Human-INTuited cues for RL, or HINTs, in which agents quickly learn to solve tasks by leveraging human coaching. Our experiments in classic control, navigation, and locomotion reveal that HINTs enables agents to learn more quickly than vision-only agents and to obtain strategies that apply to more challenging settings.

## 1 Introduction

Socio-economic factors incentivise the extension of current robotic systems to more human-centric spaces where, unfortunately, state-of-the-art robotic controllers fail due to simplifying modelling assumptions (Janner et al., 2021; Xiao et al., 2019; Deisenroth et al., 2015). Take a self-driving taxi, for example. A car equipped with onboard sensors integrates seamlessly into urban environments, but complicates state and dynamics estimation for onboard controllers because states are not fully observable. To overcome these modelling challenges, one might train an RL agent to learn a control policy directly from sensory inputs, a promising approach (Hansen et al., 2021; Schmeckpeper et al., 2021). However, learning such a policy can be data intensive, requiring countless hours of simulated experiences where the agent may learn how to extract relevant information from sensors and apply it to navigate and control the car.

On the other hand, one can look to direct supervision from good human drivers to restrict the policy search, by providing full trajectories or sub-goals that solve the task, for example. While beneficial for efficient learning, this bootstrapping approach yields policies that only succeed at task instances that are captured by human-provided data sets, requiring more samples to close the data distribution gap between the training and deployment setting. Learning visual control policies that capture complex strategies remains unsolved when we consider limits on data.

There exist many RL frameworks that leveraging human inputs for efficient skill learning, however they rely on large clean sample sets to convey a reliable learning signal. Behavioural cloning (BC) methods have been used to accomplishing feats including traversing challenging terrain, folding laundry, tying shoelaces (Peng et al., 2020; Hoque et al., 2022; Fu et al., 2024), etc. In these methods, the human typically enters the loop in a supervisory manner, providing explicit trajectories (Florence et al., 2021; Wang et al., 2023; Kumar et al., 2022) or exemplar sequences (Eysenbach et al., 2021) that encode a task solution. Agents learn faster with BC, but they do not learn far beyond demonstrated behaviours, making unfit for real-world deployment where environments and task structure are subject to variation. As the task horizon increases, such variations may make the cost of collecting ample high-quality demonstrations unjustifiable. Some approaches leverage language instructions to condition RL agents on information that guides them through phases of long

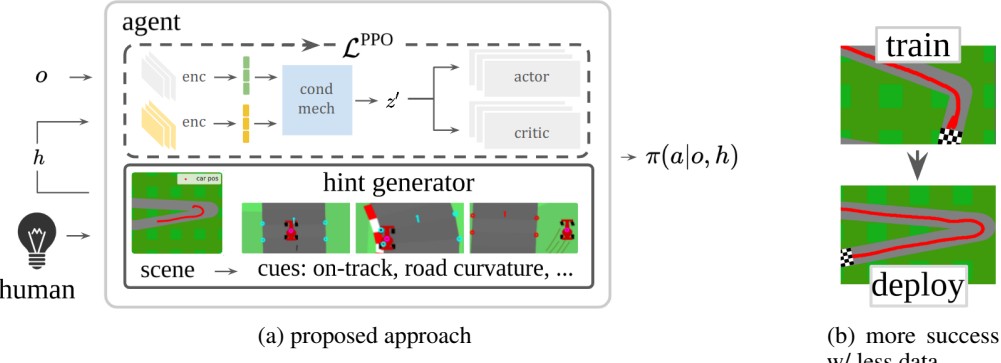

(a) proposed approach

(b) more success w/ less data

Figure 1: Our proposed framework, HINTs, addresses RL for continuous control in partially observable environments. It allows humans to coach agents while they learn from a limited number of training episodes. a) A programmatic generator $G$ (left) outputs human-intuited cues $h$ on which policies condition their actions; e.g., the road curvature. b) As a result, hint-conditioned agents learn precise control tasks with a small training budget. For more complex tasks (right), HINTs allows agents to learn more effective policies than those of vanilla agents; e.g., for passing hairpin corners.

horizon tasks (Nath et al., 2024; Hu & Clune, 2023). For control tasks, however, some strategies may be difficult to convey with exemplary data or language instruction. In the taxi example, teaching an agent to carefully applying the accelerator to avoid spinning the car out during an emergency maneuver would require many demonstrations or extensive language descriptions to cover the breadth of real-world scenarios.

As an alternative, we propose a novel way to incorporate the human into the RL training loop to guide learning without restricting the policy to prescribed solutions. Our approach allows the human serve as a coach, providing tips that help the policy learn quickly and to converge to more widely applicable strategies. For example, when teaching student drivers to drive around corners, a coach may *hint* to the student to look far into the corner. Based on this tip, the student might learn to estimate corner angles, or other *(e.g., sensory) cues* that correlation with cornering speed, over their driving experience.

Similarly, our approach, introduced in Sec 2-3, gives human experts a means of conveying concepts that guide the system toward effective control strategies, freeing them from the onus of providing clean, complete training examples. Our experiments test this idea in a classic control setting as well as a simulated car domain (Sec 4). We find that agents benefit from human coaching, realising performance improvements of +80% in classic control and +30% on transferring to challenging navigation and control tasks under tight training budgets (Sec 5).

**Contributions**   This paper presents a novel framework called HINTs, or Human-INTuited cues for RL, that allows agents to train under conceptual guidance rather than direct supervision from expert humans. We make the following key contributions:

**1)** propose a novel way to situate humans in the learning loop; namely, through a RL framework for trains policies conditioned on programmatically generated human-intuited cues,
**2)** demonstrate HINTs using multiple conditioning schemes that allow the framework to be adapted for control settings with differing visual complexity,
**3)** empirically validate this framework across a suite of visual continuous control tasks, showing dominant performance of hint-conditioned agents over state-of-the-art baselines, and
**4)** showcase that hint-conditioned agents learn more reliable strategies than purely visual agents in a uniquely challenging navigation and control benchmark.

## 2 RL FOR VISION-BASED CONTINUOUS CONTROL

Consider the problem of learning, by trial-and-error, continuous control policies from high-dimensional inputs. We formulate this problem as a partially observable Markov Decision process $\mathcal{M} = \{\mathcal{S}, \mathcal{O}, \mathcal{A}, \mathcal{P}, \rho_0, \rho, r, \gamma\}$ in which continuous states $s \in \mathcal{S} \subseteq \mathbb{R}^n$ undergo stochastic transitions given by unknown distribution $\mathcal{P}(s'|a, s)$. An agent $\pi$ generates continuous actions $a \in \mathcal{A} \subseteq \mathbb{R}^m$ given observations $o \in \mathcal{O} \subseteq \mathbb{R}^{W \times W \times c}$ from conditional distribution $o \sim \rho(o|s)$, also unspecified. The agent perceives and interacts with its environment to make progress on a task, as measured by finite reward $r(o, a) \in \mathbb{R}$. To solve a task, the agent seeks to maximise the discounted return $R_H = \mathbb{E}_{o_0 \sim \rho_0} \sum_{t=0}^{H-1} \gamma^t r_t(o_t, a_t)$ over a horizon $H$ and with $\gamma \in [0, 1)$.

In this paper, we consider an extension of this problem where a human guides learning via a known process $G(h|s, o)$ that provides the agent with a set of cues $h \in \mathcal{H} \subseteq \mathbb{R}^d$. The cues ground conceptual hints given by a human who seeks to scaffold policy learning (Guzdial, 1994). Rather than revealing a solution, these concepts convey information about important states, features, action constraints, or other aspects that support solving the task. We restrict the problem by allowing the hint generator $G$ access to ground truth the scene information such as environment state and dynamics. We address this problem by learning a hint-conditioned policy under a conditioning scheme $e$ that takes grounded hints from $G$.

## 3 LEARNING VISUAL POLICIES WITH HINTS

HINTs is a framework that views humans as coaches who can guide learning of continuous control tasks by scaffolding the learning process rather than directly supervising actions. In this framework, a knowledgeable human identifies conceptual hints that capture an intuitive strategy for solving a task. The hints guide an agent equipped with a programmatic generator $G$ which grounds conceptual hints in the scene as cues (e.g., sensory). The agent's conditional policy (Sec 3.1) receives cues generated by $G$ (Sec 3.3) on which it conditions its actions (Sec 3.2). The agent optimises a policy gradient objective using Proximal Policy Optimisation (PPO) algorithm (Schulman et al., 2017) to update its neural components. More detailed descriptions of agent specification and hint generator are covered in Appx Sec A.3.

### 3.1 AGENT SPECIFICATION

**Architecture** We choose an actor-critic agent as the substrate for our framework. The agent takes observations $o$ and hints $h$ and generates continuous actions and value function estimates. As the agent operates on image observations, we incorporate an image encoder $\pi_{\text{enc}}(o; \phi)$ parameterised by a three layer convolutional neural network with parameters $\phi$. Inputs to the encoder are normalised and centered, yielding pixels $(u, v) \in [-1, 1]^3$. The encoder outputs a latent vector $z \in \mathbb{R}^k$ which is given as input to the actor-critic components. Both the actor and critic are parameterised by three layer multilayer perceptrons (MLPs), also with parameters $\phi$. The critic outputs value estimates $V(z') = \pi_{\text{critic}}(z'; \phi)$. We implement the actor as a Gaussian policy with learnt parameters $\mu, \sigma$ which generates continuous actions $a \sim \pi_{\text{actor}}(a|z'; \phi)$. Lastly, the agent conditions its actions and value estimates on programmatically generated (grounded) hints $h$ using one of several mechanisms to generate $z'$.

**Training** Our agents learn in an on-policy fashion via policy gradient optimsation over batched experience consisting of observations, hints, actions, rewards, and termination flags. During experience collection, the agent produces cues (grounded hints via $G$) at every time step. The cues are passed to a conditioning mechanism which forwards the output to the actor-critic components. Please refer to Alg 1 for details.

### 3.2 CONDITIONING SCHEMES

Our proposed agent employs a conditioning scheme $e_\phi(h)$ to take grounded hints from humans (Fig 5). Prior to conditioning, the raw cue vectors undergo one of two transformations: a linear transformation $g_{(\cdot)} : \mathcal{H} \rightarrow \mathbb{R}^{(\cdot)}$ or a vector-to-mask $\text{mask}(h)$ where each channel contains the $i$th

---

**Algorithm 1** Training with HINTs.

---

1: Set training budget $N$, batch size $B$
2: Initialise agent: $\phi \sim \Phi$, $\pi_\phi = \{\pi_{\text{enc}}(\cdot; \phi), \pi_{\text{actor}}(\cdot; \phi), \pi_{\text{critic}}(\cdot; \phi), e_\phi(\cdot)\}$
3: Initialise environment: $E \sim \mathcal{E}$
4: **for** iter $t$ in $[1, \ldots, N]$ **do**
5: $\quad \{(o_t, a_t, r_t)\} \sim \texttt{CollectExperience}(E, \pi_\phi, B)$
6: $\quad z_t = \pi_{\text{enc}}(o_t)$, $h_t = \texttt{GenerateHint}(o_t, S_t)$
7: $\quad z' \leftarrow$ hint-conditioning using a scheme from Eqn 1-4
8: $\quad$ Calculate $v_t = \pi_{\text{critic}}(z')$, $v^* = A_t^\pi + v_{t-1}$ and $A_t^\pi$ with $\texttt{ComputeGAE}(r_{t-1}, v_{t-1})$
9: $\quad$ Calculate $\mathcal{L}^{\text{PPO}}(A_t^\pi, \pi_t, \pi_{t-1}, v^*, v_t)$ according to Eqn 5
10: $\quad$ Update $\pi_\phi$ to minimise $\mathcal{L}^{\text{PPO}}$
11: **end for**

---

dimension in the $d$-dimensional hint. These components are trained jointly using PPO algorithm. We present several designs based on the task complexity and native input size.

$$e_\phi(h) = [z, g(h; \phi)] \qquad (1)$$
$$e_\phi(h) = z + g(h; \phi) \qquad (2)$$
$$e_\phi(h) = z \cdot \alpha(h; \phi) + \beta(h; \phi) \qquad (3)$$
$$e_\phi(h) = \pi_{\text{enc}}([o, \text{mask}(h)]; \phi) \qquad (4)$$

**Latent conditioning (LC):** a simple form of conditioning where hints undergo concatenation with the image encoder's output (Eqn. 1). We apply $g_l$ before concatenating the hint to $z$. The final output $z' \in \mathbb{R}^{k+l}$ is passed to actor-critic networks $\pi_{\text{actor}}(a|z')$ and $\pi_{\text{critic}}(z')$.

**Latent additive conditioning (AC):** grounded hints are linearly combined (Eqn. 2) with the image embedding before being passed to the actor-critic networks. The hints are transformed by $g_k$ beforehand to match the dimensionality of the latent variable, yielding $z' \in \mathbb{R}^k$.

**Global feature-wise conditioning (FC):** a relaxation of the architecture proposed by Perez et al. (2017), we learn affine parameters $\alpha, \beta$ that modulate feature maps towards the final layers of the encoder block. Unique to our scheme, the feature modulator operates over embeddings rather than feature maps (Eqn. 3). If the learnt embedding space encodes semantics, then this scheme provides a way to index into encoded concepts as a function of $h$.

**Masked conditioning (MC):** a hint is concatenated with the observation before being passed to the image encoder (Eqn. 4) similar to Radford et al. (2016) and Mirza & Osindero (2014). We generate a $d$-channel mask for the $d$-dimensional hint. The result is an image in $\mathbb{R}^{W \times W \times (c+d)}$.

### 3.3 HINT GENERATOR

**Conceptual hints (from human coach)**  To address challenges in learning from image observations, we look to humans to scaffold, rather than bootstrap, learning with hints. The process of conceptualising hints in our framework requires the human to understand effective strategies for accomplishing the task. Given their intuition, the human can identify structured information that correlates with task progress across a wide variety of instances. Suppose the agent needs to solve a balancing task. A natural way to conceptualise balancing, in the human's mind, is to track swinging motion which can be grounded in state features like goal-distance and (angular or linear) velocity. Providing an agent with such cues can help shape an its internal representation of the environment or its exploration strategy. Other concepts might be declarative or process-based (e.g., providing intermediate targets). In this paper, we restrict our focus to agent performance given a fixed set of systematically available hints, enabling a richer analysis of how different conceptual hints affect learning outcomes.

**Grounded hints (cues from agent)**  Once a human identifies a conceptual hint, the generator $G$ observes the scene and computes a grounded hint $h \in \mathbb{R}^d$. We used a programmatic generator which has access to the underlying state and dynamics of the scene. This design choice made it possible to do systematic analysis of agent performance and information characteristics of $h$. Our framework does not depend on the availability of ground truth scene information to benefit learning.

| $d\theta_i/dt$, goal distance | $d\theta_i/dt$, goal distance | velocity, $\Delta x$, $\Delta z$ | on/off, curvature, speed |

Figure 2: Visualisations of grounded hints (cues). Cues computed by $G$ are illustrated and overlaid ontoto observations for a variety of environments. For example, in swing-up tasks (left), $G$ computed vertical distance from the upright position and angular velocity. These illustrations are visual aids for readers.

Fig 2 shows visualisations of cues overlaid onto image observations for a variety of environments. For swing-up and balancing tasks with $L$ links, $G$ computes state-based cues such as $D_{\text{goal}} = L\cos(\theta^*) - \sum_{i=1}^{L}\cos(\theta_i)$; illustrated by a red arrow from the target height to the vertical position of the robot. Locomotion cues consisted of subsets of state information as well as action constraints. In the driving domain, $G$ localises the car hull on the track and provides a flag $\ell$ indicating whether the car is on-track (marked by cyan/red dots) and a measure of curvature $\kappa$ (marked by forward facing vector). A full description of hints and corresponding cues are available in Tab 6 (Appx Sec A.3).

## 4 EXPERIMENTS

Our experiments evaluate agents on challenging image-based tasks in classic control, navigation and control domains, and locomotion (Towers et al., 2024; Brockman et al., 2016). We seek to understand the performance characteristics of hint-conditioned agents in two respects: 1) against those of state-of-the-art baselines and 2) as a function of hint information. Also of interest are the relative benefits of different conditioning mechanisms and how HINTS scales to more complex settings with tricky dynamics, large action spaces, and long horizons. The following hypotheses motivate our experiment design:

*H1)* under hint conditioning, agents learn faster and more proficiently than with vision alone,
*H2)* hinting with human-identified info increases performance over other types of info,
*H3)* human-identified hints can enable agents to learn more adaptive strategies than without,
*H4)* guiding with HINTS allows agents to learn quicker in high-dim action spaces than without.

**Metrics** We validate these hypotheses according to raw (train/eval) average reward computed over multiple seeds, raw average (deploy) reward as evaluation metrics, and the number of training samples.

### 4.1 AGENT SETUP

This section summarises the implementation of hint-conditioned agents and benchmark agents studied in this work. We group the baselines into three tiers shown in Tab 1. T2-3 are meant to represent upper-bounds of performance based on observability and training budget. There are several HINTS agents that correspond to different conditioning mechanisms.

**Baseline agents** We benchmark against one state-based agent and three different visual agents to benchmark the quality of grounded hints. First, we designed a vision-only agent (PPO-RGB) with the same architectural components listed in Sec 3.1, excepting the parameters associated with

Table 1: Benchmarks including widely used methods for RL, imitation learning, and Inverse RL.

| Limited training budget | | No training buget |
|---|---|---|
| vision-based | state-based | |
| PPO-RGB learns from visual-only | PPO-x learns from state-only | Expert PPO learns until convergence |
| HINTS-x is cond'd with state x | PPO-x+$\mathcal{N}(0,1)$ receives noisy state | GAIL learns by fitting to expert data |
| | HINTS-blind doesn't take image input | DAGGER an interactive IL framework |
| Tier 1 (T1) | Tier 2 (T2) | Tier 3 (T3) |

conditioning. We also implemented a state-based analogue (PPO-x) of the vision-only agent which consisted of the same architecture and hyper-parameters, excepting the input encoder. Next, we implemented a blind agent $\pi(a|g_l(h))$ (HINTs-blind) that takes grounded hints alone as input. Finally, we produced state-conditioned agents $\pi(a|o, s)$ (HINTs-x) for each scheme described in Sec 3.2. Further details about agent architecture and hyper-parameters are available in Appx Sec A.3.

To contextualise the performance of HINTs in the literature, we compared against imitation learning agents and competing model-free RL agents. To establish an upper-bound on task performance, we selected three strong state-based agents: Expert PPO (Raffin et al., 2021), DAGGER (Ross et al., 2011) and GAIL(Ho & Ermon, 2016). Additionally, they allow us to benchmark against agents that learn from direct human supervision. All of these agents were trained to convergence.

**Hint-conditioned agents** These agents are designed according to the specification in Sec 3.1 and they were *trained with the exact same hyper-parameters as* PPO-RGB. The following labels refer to agents conditioned using the mechanisms in Sec 3.2: HINTs-LC, HINTs-AC, HINTs-FC, HINTs-MC. For brevity, we eliminate HINTs prefix in the agent labels in plots. Tab 6 summarises all conceptual and grounded hints for each control task.

## 4.2 ENVIRONMENT SETUP

Each task consists of a distribution of environments $\mathcal{E}$ which have randomly initialised parameters like the agent's initial configuration, the goal configuration, physical parameters, etc. Agents are trained with a restricted training budget and tested in two regimes: standard task setting and challenging variations. The standard setting includes Classic Control tasks; e.g., Pendulum Swingup, Acrobot, and InvertedDoublePendulum. In the challenging variations, agents are trained on the standard task ($\star$) and then deployed to challenging task instances ($\star\star\star$). These instances may exhibit a different environment distribution than seen at training time. The following are brief descriptions of each domain implemented using Gymnasium (Towers et al., 2024; Brockman et al., 2016). More detailed descriptions are available in Appx Sec A.2.

$\star$ **Car Racing** Reasoning over partial state information is an unavoidable challenge agents face in this task. An agent drives a car around a winding race track while observing only its immediate surroundings. It applies gas, brake, and steering commands to control the car, carefully selecting commands so as not to sbin due to the car's power train layout. The agent achieves the goal by completing a lap as fast as possible while keeping the car on track, stabilising the car on straights and around corners.

$\star\star$ **Locomotion** To test how HINTs scales with environment complexity, we trained agents in image-based versions of MuJoCo environments with high-dimensional action spaces, focussing on locomotion (Ant, Cheetah, Humanoid). The robots are initialised in favourable configurations (e.g., standing up for locomotion) with slight perturbations to the joint states and velocities. Agents make progress on locomotion tasks by maximising forward velocity while maintaining, in some cases, a 'healthy' posture. Agents were granted 1k training episodes for Cheetah 5k for Ant, and 10k for Humanoid.

$\star\star\star$ **Challenging Variations** Agents are *trained on the standard* task and then *deployed to challenging* task instances. These instances may exhibit a different environment distribution than seen at training time. Focussing on Pendulum and Car Racing, we manipulated the environment distributions to present agents with unique challenges that test agents' ability to learn strategies that adapt to rare or less probable situations seen at training time.

*1) Pendulum Swingto:* Agents are trained on Pendulum Swingup, where the goal is fixed at the position $\theta^* = 0$. We extend the range of target positions for the swinging task to $[-\pi/2, \pi/2]$ for the deployment setting. The reward is updated to $R(\theta) = \sum_{t=1:H} \log(||\theta_t| - |\theta^*||)$ to accommodate the variety of goal states.

*2) Car Racing Hairpin:* As before, the agent is tasked with completing a lap in the shortest time, but in this case we systematically introduce a hairpin corner with high probability. This track layout was programmatically generated to produce a sharp first corner (at most 45-degrees) at every random

instance. As a consequence of the generative process, the hairpin layouts typically have, on average, less tiles than the standard track - a relevant detail for comparing the raw rewards.

## 5 RESULTS

This section enumerates several observations (tagged with *O#*) in support of our hypotheses that we will discuss in Sec 6. Across many benchmarks, HINTs showed dominant performance over agents in T1, achieving higher rewards with dramatically fewer training samples. Performance was comparable or better for T2 baselines, which have full state-information. In challenging deployment settings and Humanoid, HINTs met the performance of DAGGER.

### 5.1 CLASSIC CONTROL

**Key finding** Guiding with HINTs on a tight budget yields +75% average performance improvement.

*O1) When trained without hints, agents struggled to learn swing-up and balancing tasks.* Policies using HINTs-FC conditioning achieved large training performance gains over the PPO-RGB (Tab 4) on all tasks. The vanilla agent was unable to learn the task in the allotted training time, showing -80% performance decrease over HINTs-FC with composite hints (a combination of starred hints in Tab 4) which could solve most tasks (Tab 2). On average, the vanilla agent fell short of the strongest hint-conditioned agent by -75% across all tasks.

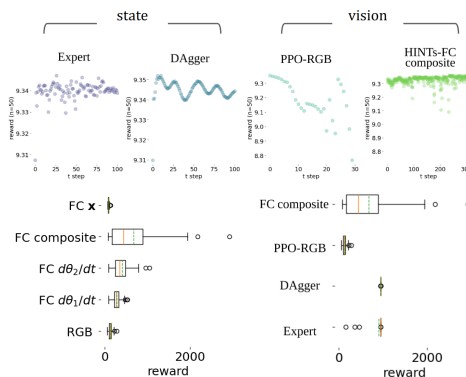

Figure 3: InvertedDoublePendulum reward over n=50 seeds. HINTs-FC outperforms PPO-RGB while approaching DAGGER.

Compared to state-based agents that were trained to convergence, hint-conditioned agents either match or approach similar performance, without demonstrations and with limited training samples. Fig 3 shows the distribution of rewards over $n = 50$ trials seen in Tab 2. Training with HINTs exceeds the performance of PPO-RGB and PPO-x while matching that of agents in T2-3. Notably, the performance of agents with weak state estimates (PPO-x+$\mathcal{N}(0, 1)$) can be overcome by HINTs with helpful human-defined hints.

### 5.2 CHALLENGING CONTROL TASKS IN LOW-DIM ACTION SPACES

**Key finding** On a challenging variation task, agents improve by +32% with composite HINTs!

⋆ **Car Racing Standard** *O2) Hint-conditioned agents with composite hints showed the highest degree of success on the standard and more challenging tracks (Tab 3).* Fig 4a demonstrates an ablation of composite hints, showing that it is necessary to condition on joint cues to excel far beyond the vanilla agent's performance. Details are available in [Appx Sec](#) A.1.

⋆⋆⋆ **Car Racing Hairpin** *O3) Tight turns proved to be a knowledge gap for vanilla agents whereas agents conditioned on hints showed higher levels of competency.* While vanilla agents demonstrated good performance in the previous setting, deploying them to tracks with extremely sharp corners, e.g., hairpins, proved challenging. Vanilla agents failed to apply the appropriate controls to make sharp turns in many cases, thereby achieving 46% average task progress. On the other hand, HINTs-MC agents conditioned on composite hints exhibited 78% average task progress. Corresponding rewards are available in Tab 3.

### 5.3 PERFORMANCE ON TASKS WITH HIGH-DIM ACTION SPACES

⋆⋆ **Ant & Humanoid** *O4)* HINTs closes performance gap to state-based and converged agents.

Tab 3 compares the average reward obtained by hint-conditioned agents to vision-only and select T2-3 agents over $n = 50$ random rollouts. Highlighted in blue are the top performing hint-conditioned agents for the locomotion domain, with Humanoid being the most challenging. PPO-x is surprisingly dominated by all hints on the Ant and Humanoid tasks. Interestingly, HINTs with speed-conditioning on Humanoid is outperformed (-22%) solely by the converged DAGGER agent. It does, however, achieve +47% on PPO-RGB.

### 5.4 PERFORMANCE VERSUS INFORMATION CONVEYED IN HINTS

**Key finding** Hinting with specific info, instead of additively more info, yields higher performance.

⋆ **Classic Control** *O5) Performance gain under hint conditioning does not linearly increase with more information.* We found that hint-conditioned agents performed significantly better with composite hints than agents given hints that conveyed full state information. In all tasks (Tab 2), performance of agents given partial or full state information performed at least -56% worse on average compare to agents given composite hints.

⋆⋆⋆ **Pendulum Swingto** *O6) Composite hints are useful for training on complex tasks, but they may not generalise as well as non-composite hints.* When tasked with reaching an arbitrary goal position, learning with composite hints yielded agents that improved by +50% over PPO-x. However, when trained on Pendulum Swingup and deployed to Swingto, only the angular velocity-conditioned agent closed the performance gap to state-based agents. The goal distance component of the composite hint may have yielded a policy that overfit to seeing a majority of training observations close to the upright position.

## 6 DISCUSSION & CONCLUSION

This paper considers visual continuous control in data-constrained learning scenarios and presents HINTs, a novel framework that efficiently trains RL agents by leveraging coaching from human experts. We showed that with hints, agents can learn precise control tasks, including challenging cases of goal reaching and car navigation. Often learning with composite hints showed the best performance, notably over HINTs-x *(H2;O5,O6)*. Hint-conditioned agents were capable of improving on average by +80% over vision-only agents in a classic control domain, training dramatically faster *(H1:O1,O2)*. In support of hypothesis *(H3:O3,O5,O6)*, we find that providing agents with a specific choice of human-identified hints (e.g., concise composites) can enable agents that learn more broadly applicable strategies. In the challenging variations domain, HINTsachieved an improvement of +32% on PPO-RGB and +50% over PPO-x. Our experiments in the locomotion domain showed a capacity for hint-conditioned agents to learn more proficiently in high-dimensional state spaces than vision-only agents *(H4;O4)*. In some cases, HINTs closes the performance gap to state-based agents that leverage millions of training samples and costly human demonstrations.

**Limitations** We have demonstrated that HINTs successfully trains agents to do challenging control tasks with the help of human coaching. However, the current implementation relies on programmatic generator $G$, which may be challenging to design for real world domains where state information is inaccessible. Further, Sec 5 suggests that some hints have relatively higher utility than others, depending on the task. But it is unclear what metrics can capture this difference so that human experts can quickly identify productive hints.

## 7 RELATED WORK

**Efficient learning through modelling and policy distillation** Many approaches overcome partial observability by leveraging state representation learning, explicit world modelling (Hafner et al., 2019), and transferring policies from rich simulation environments to the real world. Privileged information, or additional data that is not present in the deployment setting, can address challenging representation learning (Zhang et al., 2025) and skill learning (Wang et al., 2024; Mesnard et al., 2021) problems by training expert policies in stateful environments and transferring skills to policies in partially observable environments (Galashov et al., 2022; Kamienny et al., 2020; Zhang et al.,

2025). Some works have introduced distillation methods that allow an RL agent with access to privileged information to coach an imitation learning agent in an urban driving domain (Zhang et al., 2021). The RL coach provided explicit supervision over actions whereas we propose a more flexible framework where the agent can leverage conceptual guidance to learn behaviours that extend to variable deployment settings.

**Efficient learning through conditional-agents** Conditioning agents on informative language instructions and goals has been a common approach to efficiently train visual RL agents for a variety of task domains (Xu et al., 2022; Wu et al., 2025; Horita et al., 2021). Goal conditioned RL allows agents to adapt their actions based on goals prescribed by generative models (Huang et al., 2024; Hafner et al., 2022). This approach can be useful in long horizon tasks (Luu et al., 2025; Hafner et al., 2022) where sparse rewards provide weak learning objectives. However, in continuous control settings with partial observability, it may not be clear how to guide systems with goal observations. Integrating language and goal-conditioning is another option, but many approaches have focussed on higher level task domains than control; e.g., planning and reasoning (Nath et al., 2024; Hu & Clune, 2023). Recent work has explored ideas around supervising agents at a conceptual level, but these methods have largely been demonstrated on small RL problems with finite state and action spaces (Hu & Clune, 2023; Myers et al., 2023). We instead explore how conceptual guidance can aid agents in solving visual continuous control with data constraints.

Table 2: Evaluation performance of hint-conditioned agents in image-based Classic Control tasks Pendulum, Acrobot, and InvertedDoublePendulum (IDP) over n=50 random seeds. Hint-conditioned agents take RGB inputs and a variety of hints (via mechanism in Eqn 3-4). Hints with marked with $\star$ are constituents of the composite hint.

| Avg reward ↑ | No training budget | | | Tight train budget | | | | | | | | |
|---|---|---|---|---|---|---|---|---|---|---|---|---|
| Agents | Expert | DAGGER | GAIL | PPO | | | HINTs-FC | (conditioning w/ more info →) | | | | |
| | $\mathbf{x}$ | $\mathbf{x}$ | $\mathbf{x}$ | $\mathbf{x}$ | $\mathbf{x} + \mathcal{N}(0,1)$ | RGB | $\star\, d\theta_1/dt$ | $\star\, d\theta_2/dt$ | joint | composite | $\mathbf{x}_{\text{partial}}$ | $\mathbf{x}$ |
| Acro | -72.44 $\pm13.1$ | -67.30 | -500.00 $\pm0.0$ | -197.90 $\pm36.4$ | -201.60 $\pm41.1$ | -500.00 $\pm0.0$ | -212.38 $\pm52.2$ | -228.98 $\pm63.2$ | -302.24 $\pm75.7$ | -197.40 $\pm41.4$ | -241.66 $\pm46.5$ | -500.00 $\pm0.0$ |
| | | | | | | | $\star\, d\theta_1/dt$ | $\star\, d\theta_2/dt$ | $D_{\text{goal}}$ | composite | $\mathbf{x}_{\text{partial}}$ | $\mathbf{x}$ |
| IDP | 906.34 $\pm150.2$ | 943.85 $\pm0.8$ | 943.26 $\pm0.7$ | 944.67 $\pm0.2$ | 109.98 $\pm25.9$ | 130.99 $\pm41.8$ | 281.14 $\pm99.6$ | 400.34 $\pm215.6$ | 40.07 $\pm9.7$ | 680.84 $\pm659.8$ | 70.35 $\pm19.0$ | 88.56 $\pm14.4$ |
| | | | | | | | $\star\, d\theta/dt$ | $a = \tau(\theta)$ | $\star\, D_{\text{goal}}$ | composite | $\mathbf{x}_{\text{partial}}$ | $\mathbf{x}$ |
| Pend *Swingup* | -174.77 $\pm96.7$ | -253.93 $\pm216.4$ | -191.99 $\pm118.9$ | -1270.86 $\pm77.3$ | -1100.96 $\pm66.8$ | -1352.82 $\pm169.6$ | -250.31 $\pm164.0$ | -1236.58 $\pm256.8$ | -1203.08 $\pm307.6$ | -241.14 $\pm178.5$ | -1304.77 $\pm180.1$ | -1248.46 $\pm150.5$ |
| | | | | | | | $\star\, d\theta/dt$ | composite | $\mathbf{x}$ | | | |
| *Swingto deploy* | - | 111.82 $\pm137.8$ | - | 120.15 $\pm85.3$ | 115.20 $\pm44.5$ | 21.26 $\pm138.9$ | 111.08 $\pm186.4$ | -118.48 $\pm72.0$ | 18.76 $\pm105.5$ | | | |
| | | | | | | | $\star\, d\theta/dt$ | composite | $\mathbf{x}$ | | | |
| *Swingto train/eval* | - | - | - | 121.89 $\pm65.1$ | - | 22.81 $\pm130.1$ | 127.07 $\pm72.7$ | 246.14 $\pm99.1$ | 13.90 $\pm126.9$ | | | |

Table 3: Evaluation performance in complex tasks ($n$=50 random seeds). Each task is marked with shorthands MC and FC indicating whether HINTs-MC or HINTs-FC was used. Hint-conditioning outperforms the vision-only agent on all tasks. Except for Cheetah, HINTs also outperforms PPO-$\mathbf{x}$ by a large margin. Human guidance via HINTs with a variety of concepts approaches that of DAGGER, but with no demonstrations and limited training samples.

| Avg reward ↑ | No training budget | | Tight train budget | | | | |
|---|---|---|---|---|---|---|---|
| Agents | DAGGER | PPO | PPO | HINTs (w/ FC or MC; Eqn 3 or Eqn 4) | | | |
| | $\mathbf{x}$ | $\mathbf{x}$ | RGB | $\star$ curvature | $\star$ on-track | composite | |
| CR Standard *trn/evl MC* | N/A | N/A | 461.17 $\pm238.6$ | 539.27 $\pm284.5$ | 532.02 $\pm269.0$ | 590.22 $\pm261.9$ | |
| CR Hairpin *deploy MC* | N/A | N/A | 419.95 $\pm302.3$ | 479.76 $\pm296.4$ | 366.51 $\pm281.2$ | 720.45 $\pm272.1$ | |
| | | | | speed | $D_{\text{goal-height}}$ | $\mathbf{x}$ | |
| Ant (FC) | 4000.42 $\pm152.8$ | 533.77 $\pm254.7$ | 1015.49 $\pm3.3$ | 1313.57 $\pm57.6$ | 1442.82 $\pm162.6$ | 1007.52 $\pm5.1$ | |
| | | | | reward | $dx/dt$ | $\max_{<t}(dx/dt)$ | $\mathbf{x}$ |
| Cheetah (FC) | 2351.16 $\pm109.6$ | 2327.99 $\pm535.8$ | 1.82 $\pm18.6$ | 55.41 $\pm24.5$ | 138.45 $\pm19.3$ | 21.91 $\pm66.5$ | 30.42 $\pm13.2$ |
| | | | | speed | $D_{\text{goal-height}}$ | | |
| Human (FC) | 588.00 $\pm152.4$ | 228.30 $\pm17.6$ | 310.86 $\pm41.7$ | 455.95 $\pm94.5$ | 306.14 $\pm52.5$ | | |

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

## A APPENDIX

### A.1 ADDITIONAL RESULTS - HINT ABLATION & CHALLENGING VARIATIONS

**On hinting with more info**   Simply hinting with more information is not always better. Surprisingly, this trend holds for a variety of conditioning mechanisms and control tasks (Tab 4), suggesting that the observed performance differences result from differences in hint information content versus task utility. We find that conditioning on partial and full state information does not recover the performance observed with PPO-x, possibly due to imbalance in the state's signal strength versus that of the visual modality. A similar trend appears for Pendulum Swingto and locomotion tasks (Tab 2 and 3).

More precisely, Tab 4 shows that grounding the swing conceptual hint in angular velocity yields +80%, +60%, and +87% performance improvement in Pendulum Swing-up, Acrobot, and Inverted-DoublePendulum over hinting with full state information. This conceptual hint is highly effective in Pendulum and Acrobot tasks where the agent needs to generate high torque to change the angular velocity of the robot links and then stabilise. The grounded hint can be as simple as angular velocity. On the other hand, for InvertedDoublePendulum, the swinging concept is associated with a unimodal strategy, primarily to stabilise. Grounding the concept in angular velocity proves useful, though not optimal given a limited training budget.

**Challenging variation 1**   HINTs yields agents capable of negotiating hairpin corners more successfully than vision-only agents. As shown in Fig 4, agents that perform this task purely from the cues alone (blind) perform dramatically worse. However, we observe a trend across blind and conditional policies that composite hints yield dominant performance within each policy type. To gain more insight into this trend, Fig 4b shows that composite-conditioned agents achieve a higher concentration of high reward trajectories for longer horizons.

Table 4: Performance of hint-conditioned agents benefits more from utility of hints rather than high information content. Mean rewards ($n$=5 agents) for hint-conditioned agents trained in classic control environments; Pendulum, Acrobot, and InvertedDoublePendulum. Agents conditioned their policies on cues like angular velocity, goal distance, and composites (e.g., $\{d\theta/dt, D_{\text{goal}}\}$ for Pendulum). Hinting with partial state info (e.g., excluding $d\theta/dt$ in Pendulumand $x, y$ in Acrobotand InvertedDoublePendulum) yielded policies that performed comparably to PPO-RGB.

| Avg reward ↑ | Pendulum | | | | Acrobot | | | | InvertedDoublePendulum | | | |
|---|---|---|---|---|---|---|---|---|---|---|---|---|
| | $d\theta/dt$ | composite | $x_{\text{parital}}$ | $x$ | $d\theta_1/dt$ | $d\theta_2/dt$ | composite | $x$ | $d\theta_1/dt$ | $d\theta_2/dt$ | composite | $x$ |
| HINTS-FC | -2895.88 ±1767.0 | -2550.84 ±1812.2 | -6445.25 ±755.1 | -6423.87 ±597.8 | -1414.50 ±408.3 | -1703.50 ±458.0 | -1429.17 ±528.2 | -2500.00 ±0.0 | 612.94 ±165.0 | 817.53 ±403.8 | 1998.69 ±1288.0 | 443.80 ±72.9 |
| HINTS-LC | -5900.90 ±638.9 | -3801.58 ±1142.6 | -6166.61 ±1026.3 | -6419.55 ±668.3 | -2008.83 ±303.4 | -2258.17 ±254.7 | -1534.00 ±498.7 | -2499.33 ±1.5 | 592.78 ±145.2 | 753.34 ±274.7 | 1334.98 ±927.0 | 462.45 ±112.2 |
| HINTS-AC | -6194.73 ±365.9 | -5087.93 ±875.5 | -6227.93 ±907.5 | -6357.87 ±916.2 | -1922.83 ±351.4 | -2104.00 ±298.9 | -1432.50 ±536.1 | -2500.00 ±0.0 | 636.26 ±207.0 | 683.84 ±216.7 | 1690.04 ±1191.5 | 437.73 ±53.7 |
| HINTS-MC | -6473.45 ±629.3 | -6379.08 ±688.7 | -6408.13 ±818.5 | -6439.26 ±788.6 | -2363.83 ±111.0 | -1934.33 ±316.0 | -2350.00 ±155.3 | -2500.00 ±0.0 | 640.28 ±182.7 | 745.28 ±284.9 | 983.13 ±798.7 | 404.53 ±67.5 |
| | conditioning w/ more info → | | | | conditioning w/ more info → | | | | conditioning w/ more info → | | | |

## A.2 ENVIRONMENT DETAILS

**Classic Control**   We begin by validating HINTs in image-based versions of Pendulum Swingup, Acrobot, and InvertedDoublePendulum task (Towers et al., 2024; Brockman et al., 2016) where we can analyse the effect of design choices on learning outcomes. In this setting, an agent must actuate a robot to reach a goal configuration to maximise its reward. Agents observe 64-by-64 RGB renderings of the scene and a variety of hinting information (Appx Sec 6). Each task begins with a random state $x_0 = (\theta_0, \dot\theta_0)$ and a fixed goal. Except for Expert PPO, DAGGER, GAIL, and DRQV2, every agent had a training budget of 700 or 1k episodes.

**Car Racing**   Reasoning over partial state information is an unavoidable challenge agents face in this task. An agent drives a car around a winding race track while observing only its immediate surroundings. The agent sees 96-by-96 RGB snapshots of the car's location on the track from a bird's eye view. It applies gas, brake, and steering commands to control the car, carefully selecting commands so as not to sbin due to the car's power train layout. The agent achieves the goal by completing a lap as fast as possible while keeping the car on track, stabilising the car on straights and around corners.

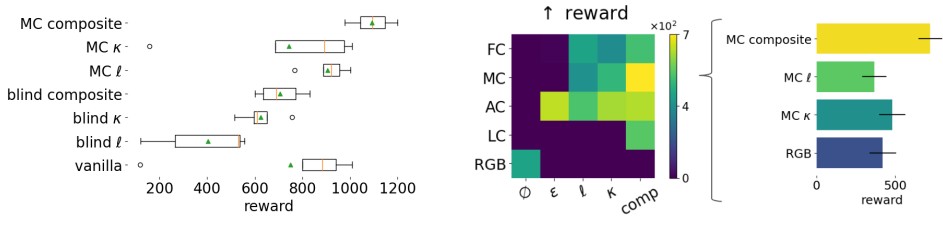

(a) Train in Car Racing Standard                    (b) Deploy in Car Racing Hairpin

Figure 4: Hint conditioning is necessary to achieve high performance. a) Training rewards achieved by HINTs-MC policy with various hints against vision-only policy and corresponding blind policies. Conditioning with composite hints shows similar dominance as in the Classic control domain. b) HINTs enables agents to learn strategies that yield +30% increase (for composite hints) in reward in the deployed Car RacingHairpin task. The constituent hints, on-track label $\ell$ and curvature $\kappa$, showed improved training performance, but curvature proved to be more useful in the deployment setting.

Figure 5: Hint conditioning mechanisms. Agents take image inputs and grounded hints programmatically generated by $G$ and output continuous actions. The inputs are processed by an image encoder and hint a encoder which produce $z$ and $h$. A conditioning mechanism $e$ to yields inputs to the actor and critic networks from $z$ and $h$. The conditioning scheme may be task dependent. For example, tasks in which the dimensionality of cues is dominated, e.g., $d << k$, conditioning at the input may preserve hint signal better than conditioning in the latent space.

Table 5: Parameters for training hint-conditioned agents and PPO-based baselines.

| | Input Space | Horizon $H$ | Batch Size $B$ | Learn Rate (l.r.) | Actor l.r. | Critic l.r. | Hint Dim $l$ |
|---|---|---|---|---|---|---|---|
| Pendulum | 3 | 1024 | 1024 | 1e-3 | 1e-3 | 1e-3 | - |
| Acrobot | 6 | 1024 | 1024 | 1e-3 | 3e-4 | 1e-3 | - |
| IDP | 9 | 1024 & 200 | 1024 | 1e-3 | 3e-4 | 1e-3 | - |
| Ant | 105 | 1024 & 200 | 1024 | 1e-3 | 3e-4 | 1e-3 | - |
| Cheetah | 17 | 1024 & 200 | 1024 | 1e-3 | 3e-4 | 1e-3 | - |
| Humanoid | 348 | 1024 & 200 | 1024 | 1e-3 | 3e-4 | 1e-3 | - |
| Classic Control | 64x64x3 | 1024 | 1024 | 1e-3 | 1e-3 & 3e-4 | 1e-3 | 64 |
| Locomotion | 64x64x3 | 1024 & 200 | 1024 | 1e-3 | 3e-4 | 1e-3 | 64 |
| Car | 96x96x3 | 1000 | 1024 | 1e-3 | 3e-4 | 1e-3 | 64 |

We gave agents a budget of 1000 training episodes, randomising the track layout for every instance. We set $H = 1000$ time steps for data collection and evaluation trials. The agent's reward is proportional to the number of visited track tiles (denoted by $n_t$). We used this value to compute a progress metric which represents track completion percentage: $P = n_t/N$, where $N$ is the track length.

**Locomotion**   To test how HINTs scales with environment complexity, we deployed agents to image-based versions of MuJoCo environments with high-dimensional action spaces (Towers et al., 2024; Brockman et al., 2016), focussing on locomotion (Ant, Cheetah, Humanoid). The robots are initialised in favourable configurations (e.g., standing up for locomotion) with slight perturbations to the joint states and velocities. Agents make progress on locomotion tasks by maximising forward velocity while maintaining, in some cases, a 'healthy' posture. Agents were granted 1k training episodes for Cheetah 5k for Ant, and 10k for Humanoid.

A.3   AGENT IMPLEMENTATION DETAILS

**Agent training**   We based the training objective for all agents on clipped surrogate objective (Eqn 7 Schulman et al. (2017)). The advantages were computed over reward batch of size 1024 and with $\gamma = 0.99$ and $\lambda_{\text{gae}} = 0.95$. The policy ratio was computed using log-probabilities of actions taken by the current $\pi_{\phi'}$ and old $\pi_{\phi}$ policies. Additionally, we regularised using a policy entropy loss term as well as a value function regression term with coefficients $\lambda_{\text{val}} = 0.5, \lambda_{\text{ent}} = 0.01$. Hint-conditioned agents were trained using Algorithm 1.

$$\mathcal{L}^{\text{PPO}} = \mathcal{L}^{\text{CLIP}}(A^{\pi}, \pi_{\phi}, \pi_{\phi'}) + \lambda_{\text{ent}}\mathrm{H}(\pi_{\phi}) + \lambda_{\text{val}}\mathrm{MSE}(v^*, v) \tag{5}$$

**Algorithm 2** Training with HINTs (not-compact).

1: Set training budget $N$, batch size $B$
2: Initialise agent: $\phi \sim \Phi, \pi_\phi = \{\pi_{\text{enc}}(\cdot; \phi), \pi_{\text{actor}}(\cdot; \phi), \pi_{\text{critic}}(\cdot; \phi), e_\phi(\cdot)\}$
3: Initialise environment: $E \sim \mathcal{E}$ $D_B = \texttt{CollectExperience}(E, \pi_\phi, B)$
4: $\{(o_0, a_0, r_0)\} \sim D_B$
5: **for** iter $t$ in $[1, \ldots, N]$ **do**
6: $\quad \{(o_t, a_t, r_t)\} \sim D_B$, $S_t = \texttt{SceneInfo}(E)$
7: $\quad z_t = \pi_{\text{enc}}(o_t)$, $h_t = \texttt{GenerateHint}(o_t, S_t)$
8: $\quad$ Apply hint-conditioning (Fig 5) using a scheme from Eqn 1-4
9:
10: $\quad$ Calculate value estimate and target as $v_t = \pi_{\text{critic}}(z')$, $v^* = A_t^\pi + v_{t-1}$
11: $\quad$ Compute advantage $A_t^\pi$ with $\texttt{ComputeGAE}(r_{t-1}, v_{t-1})$
12: $\quad$ Calculate $\mathcal{L}^{\text{PPO}}(A_t^\pi, \pi_t, \pi_{t-1}, v^*, v_t)$ according to Eqn 5
13:
14: $\quad$ Update $\pi_\phi$ to minimise $\mathcal{L}^{\text{PPO}}$
15: $\quad D_B = \texttt{CollectExperience}(E, \pi_\phi, B)$
16: **end for**

Table 6: A log of all conceptual and grounded hints for each environment presented in Sec 5.

| Environment | Hint | | | | | |
|---|---|---|---|---|---|---|
| | | reach goal | | swing | | None |
| Pendulum | goal distance | composite | angular velocity | torque | partial-state | state |
| | $D_{\text{goal}}$ | $\{d\theta/dt, D_{\text{goal}}\}$ | $d\theta/dt$ | $a = \tau(\theta)$ | $\mathbf{x}[:2]$ | $[x, y, d\theta/dt]$ |
| | composite | $i$-angular velocity | joint | partial-state | | |
| Acrobot | $\{d\theta/dt, D_{\text{goal}}\}$ | $d\theta_i/dt$ | $\sum_i d\theta_i/dt$ | $\mathbf{x}[2:]$ | $\mathbf{x}$ | |
| | $D_{\text{goal}}$ | composite | $i$-angular velocity | partial-state | | |
| IDP | $\sum_i d\theta_i/dt$ | $\{d\theta/dt, D_{\text{goal}}\}$ | $d\theta_i/dt$ | $\mathbf{x}[2:]$ | $\mathbf{x}$ | |
| | | use corner speed | | go slow | | |
| Car Racing | speed | curvature | | composite | on-track | |
| | $\sqrt{dx/dt^2 + dy/dt^2}$ | $\kappa$ | | $\{\ell, \kappa\}$ | $\ell$ | |
| | | move along | stay healthy | None | | |
| Ant | speed | goal height | state | | | |
| | $\sqrt{dx/dt^2 + dy/dt^2}$ | $D_{\text{goal-height}}$ or $D_{z^*}$ | $\mathbf{x}$ | | | |
| | | keep move along | None | | | |
| Cheetah | velocity | max-seen-velocity | state | | | |
| | $dx/dt$ | $\max_{<t}(dx/dt)$ | $\mathbf{x}$ | | | |
| | | move along | stay healthy | None | | |
| Humanoid | velocity | goal height | state | | | |
| | $dx/dt$ | $D_{\text{goal-height}}$ | $\mathbf{x}$ | | | |