# OpenReview forum: "HINTs: Human-INTuited Cues for Reinforcement Learning"
_ICLR.cc/2026/Conference — Submitted to ICLR 2026_

### Official Review · Reviewer_o8SK · 2025-10-31

**Soundness:** 3
**Presentation:** 2
**Contribution:** 3
**Rating:** 2
**Confidence:** 4

**Summary:**

This paper provides a framework for allowing humans to give observations to an RL agent (human in the loop) during training to improve sample efficiency. The high level idea is to show the agent where a reasonable location to focus attention is with limited samples. They provide four options for giving their agents clues in the space of image inputs. They evaluate their method on a set of vision-based RL experiments.

**Strengths:**

The general idea of giving agents feedback is interesting and relevant.

The clarity of the paper is moderate. The general idea is described well.

The significance of this work is good. Improving RL-agents by adding humans in the loop is a well motivated idea and this is a good method for it.

**Weaknesses:**

The originality of the work seems to be reasonable but I think there are missing a section in their related work about human in the loop RL which seems to be the most relevant part of the literature (https://jair.org/index.php/jair/article/view/15348).

In the empirical results the tables (2 and 3) are confusing. I like the confidence intervals (thank you for that) but it is very difficult to tell what is going on. Specifically, it is hard to understand what each of the HINT-FCs changes and why they are different. As well why are these figures not in the empirical section where you discuss them afterward and instead at the very end? Figure 3 is tiny. I can't tell what is going on at all here at all.

There is no description of the limitations of this method. There is no conclusion or general discussion. You should have a discussion about weaknesses and a conclusion not just end randomly.

The generator implementation isn't described very well, specifically the section 3.3 seems critical but isn't very clear.

I think with added clarifications this is a good paper and I'll gladly raise my score if these are addressed.

**Questions:**

051 - Agents learn faster? In what sense? Time?

041 - Can just use the word "in distribution"

142 - The agent takes in hints as observations. How does it work without hints during inference?

220 - How do we make sure the agent doesn't over rely on hints?

330 - So a problem with HIL RL is that it isn't possible for the real world to just "stop" and give hints. Is there a way to incorporate this into off policy or offline RL instead?

---

> ### Author Response · Authors · 2025-11-21
>
> Thank you for your positive evaluation of our paper. We are appreciative of your suggestions to improve the clarity of our presentation. Below are our responses to your questions and comments:
>
>
> __positioning of our work__
>
>
> >*I think there are missing a section in their related work about human in the loop RL*
>
>
> We have extended our related work section to include the following:
>
>
> Human-in-the-loop RL [[Najar+21](https://www.frontiersin.org/journals/robotics-and-ai/articles/10.3389/frobt.2021.584075/full),[[Retzlaff+24](https://dl.acm.org/doi/10.1613/jair.1.15348)] incorporates human input into the learning process so that an agent can satisfy multiple objectives (e.g., efficiency, safety, adaptability, etc.). This paradigm has at least two schools of thought in which the human interacts dynamically with the agent or supervises the agent in an offline sense. Approaches with an iterative dynamic use human input to shape policies [[Knox+09](https://www.cs.utexas.edu/~bradknox/papers/kcap09-knox.pdf),[Ross+11](https://arxiv.org/abs/1011.0686)], estimate reward functions and value [[Christiano+17](https://arxiv.org/abs/1706.03741),[Liu+19](https://openaccess.thecvf.com/content_ICCV_2019/html/Liu_Deep_Reinforcement_Active_Learning_for_Human-in-the-Loop_Person_Re-Identification_ICCV_2019_paper.html),[Guan+21](https://papers.nips.cc/paper_files/paper/2021/file/b6f8dc086b2d60c5856e4ff517060392-Paper.pdf),[Ouyang+22](https://arxiv.org/abs/2203.02155),[MacGlashan+23](https://arxiv.org/abs/1701.06049)], and more [[Abel+17](https://arxiv.org/abs/1701.04079)]. On the other hand, human input can be integrated into learning without repeated interactions, as in reward inference from preferences, demonstrations [[Finn+16](https://proceedings.mlr.press/v48/finn16.pdf),[Szot+23](https://arxiv.org/abs/2303.16194),[Kim+23](https://proceedings.neurips.cc/paper_files/paper/2023/file/124dde499d62b58e97e42a45b26d7369-Paper-Conference.pdf)], or suboptimal data [[Muslimani+25](https://arxiv.org/abs/2405.00746)]. In our approach, humans can target more than the reward function through the hint generator, contrary to the latter approaches. We thereby minimise the need for humans to provide expert demonstrations or accurate preferences by requesting conceptual hints which the generator grounds in the environment.
>
>
>
>
>
>
> __results and conclusion__
>
>
> > *In the empirical results the tables (2 and 3) are confusing*
>
>
> We acknowledge this point and appreciate your specificity. Please refer to [our initial response](https://openreview.net/forum?id=W0tpFxWcdd&noteId=eLevvckNTP) above where we have included an example of a revised table and caption. We have also reduced Figure 3 which elaborates on the tabulated results for InvertedDoublePendulum in Table 2. We are happy to continue iterating on the formatting to improve clarity.
>
>
> > *There is no description of the limitations of this method. There is no conclusion or general discussion.*
>
>
> Section 6 contains the discussion, limitations, and conclusion. Conventionally, this is the final section of most papers instead of related work, as we have in our paper. It would clarify the order if we stated the following at the end of the introduction:
> "Similarly, our approach, introduced in Sec 2-3 ... We contextualise these findings in the literature (Sec 7) and conclude with insights and limitations based on the results in Section 6."
>
>
> __wording__
>
>
> > *Can just use the word "in distribution"*
>
>
> We have revised accordingly: "While beneficial for efficient learning, this bootstrapping approach yields policies that only succeed at in-distribution task instances"...
>
>
>
>
> > *The generator implementation isn't described very well (section 3.3)* [@o8SK]
>
>
>
>
> Thank you for highlighting the need for clarity on this point. We have expanded on the generator description to discuss how this dependency is broken via a learnt generator. __Please review https://github.com/charlie-the-brave/anon-hints/blob/reviews/reformat/revisions.pdf__.
>
>
> ## references
>
>
> Retzlaff et al. Human-in-the-Loop Reinforcement Learning: A Survey and Position on Requirements, Challenges, and Opportunities. JAIR 2024.
>
>
> Najar et al. Reinforcement Learning With Human Advice: A Survey. FRAI 2021.
>
>
>
> Muslimani et al. Leveraging Sub-Optimal Data For Human-Inthe-Loop Reinforcement Learning. ICLR 2025.
>
>
>
>
> Szot et al. BC-IRL: Learning Generalizable Reward Functions From Demonstrations. ICLR 2023.
>
>
>
>
> Kim et al. Learning Shared Safety Constraints from Multi-task Demonstrations. NIPS 2023.
>
>
>
>
> Ouyang et al. Training language models to follow instructions with human feedback. 2022.
>
>
>
>
> Guan et al. Reinforcement Learning with Explanation and Context-Aware Data Augmentation. NIPS 2021.
>
>
>
> Finn et al.  Guided Cost Learning: Deep Inverse Optimal Control via Policy Optimization. 2016.
>
>
> MacGlashan et al. Interactive Learning from Policy-Dependent Human Feedback ICML 2017
>
> __full list in pdf__

---

### Official Review · Reviewer_rFfh · 2025-11-01

**Soundness:** 3
**Presentation:** 2
**Contribution:** 1
**Rating:** 2
**Confidence:** 3

**Summary:**

This paper addresses the challenge of data-efficient learning from pixels in continuous-control reinforcement learning. It introduces HINTS, a framework where human-designed conceptual hints (such as curvature or orientation) are grounded into numerical cues and provided to the policy through lightweight conditioning schemes. HINTS allows policies to integrate structured guidance with visual input, improving learning speed and stability. Experiments on standard benchmark tasks show that HINTS significantly outperforms vision-only baselines and often matches state-based agents under the same sample budget.

**Strengths:**

1. **Conceptually simple and practical.** The core idea is straightforward yet useful in practice—real-world agents can often benefit from human-provided hints or structured cues, making the approach both intuitive and implementable.
1. **Appropriate empirical design.** Results are averaged over 50 random seeds, providing good statistical confidence; I appreciate this, as many papers report results with far fewer runs.
1.** Well-structured presentation.** The paper is well signposted, with each experimental subsection beginning with a clear takeaway that makes the narrative easy to follow and the results easier to interpret.

**Weaknesses:**

1. **Clarify the generator and hint definitions.** The paper provides intuitive explanations for the generator and hints, but is unclear what exactly they are at the implementation level. My understanding is that the generator is a hardcoded rule that maps environment states to handcrafted features (the hints). Please clarify how G is implemented in each domain and what form the hint vectors take in practice. If the hint is just a handcrafted feature, then HINTs is simply bypassing the challenge of learning from pixels by having a human provide some useful features so the agent doesn't have to learn features on its own.
1. **Weak Results.** Results in Tables 2 and 3 look statistically insignificant. I'm not sure what if the +/- uncertainty represents standard error, 95% confidence interval, etc., but the uncertainty intervals for HINTs are quite large and often overlap with the uncertainty intervals of other methods.
1. **Ground-truth dependency.** The paper claims that HINTS “does not depend on the availability of ground-truth scene information,” yet all reported experiments appear to rely on privileged state access to compute the hints. It would strengthen the paper to include at least one setting where hints are inferred without using ground-truth data.

1. **Presentation and layout issues.** Many tables and figures appear far from their in-text references and are sometimes cited out of order, which disrupts the reading flow. Please reorder or reposition them closer to the discussion paragraphs. In addition, several figures contain text that is very small and difficult to read; enlarging labels or simplifying legends would improve clarity.

Two comments:

* > Similarly, our approach, introduced in Sec 2-3, gives human experts a means of conveying concepts that guide the system toward effective control strategies, freeing them from the onus of providing clean, complete training examples.

    This idea is reminiscent of Guided Data Augmentation (GuDA) [1], which generates expert-quality augmented data by leveraging human intuition about what constitutes task progress. In GuDA, the human is not required to *generate* expert actions directly but only to *identify* whether a given trajectory exhibits expert-level behavior.

* I think it's worth explicitly mentioning that it is reasonable to expect a human to identify or design such cues in real-world tasks. Part of the LR community really dislikes priors like this, but I say it's a very reasonable expectation for the targeted problem settings.


[1] Corrado et. al. Guided Data Augmentation for Online Reinforcement Learning and Imitation Learning. RLC 2024. https://arxiv.org/abs/2310.18247

**Questions:**

1. Is the generator a hardcoded, task-specific rule? Is the hint just a handcrafted feature?
2. Can the authors comment on the statistical significance of the results in Tables 2 and 3?

---

> ### Author Response · Authors · 2025-11-21
>
> Thank you for your constructive points. We are glad to hear that our presentation was effective, but we will continue to refine according to your suggestions. Below we address your feedback:
>
> __implementation details__
>
> > 1. *Is the generator a hardcoded, task-specific rule? Is the hint just a handcrafted feature?*
>
> Our implementation includes both a programmatic and a learnt generator. The grounded hints are not just hand-crafted features, though we acknowledge the lack of clarity. Please see [our initial response](https://openreview.net/forum?id=W0tpFxWcdd&noteId=eLevvckNTP) that addresses this question in more detail.
>
> Many of our results use state-based groundings. However, we found it beneficial to analyse the benefits of guidance with interpretable cues. In the pendulum experiments, for example, we found that noise-based hints, possibly as a form of regularisation, can affect performance positively.
>
>
> >[@rFfh]
> >
> > 1. *Clarify the generator and hint definitions... unclear what exactly they are at the implementation level*
> >
> > 3. *Ground-truth dependency. The paper claims that HINTS “does not depend on the availability of ground-truth scene information,” yet all reported experiments appear to rely on privileged state access to compute the hints*
> >
> >  [@o8SK]
> >
> > *The generator implementation isn't described very well (section 3.3)*
>
> Thank you for highlighting the need for clarity on this point. We have expanded on the generator description to discuss how this dependency is broken via a learnt generator. __Please see https://github.com/charlie-the-brave/anon-hints/blob/reviews/reformat/revisions.pdf__.
>
> With the learnt generator, learning happens in two phases, 1) the generator learns from observations to solve a regression task in simulation prior to 2) policy learning under the generated cues.
>
> __results and significance__
>
> > 2. *Weak Results.  What does  +/- uncertainty represent? HINTs statistical significance?*
> >
> > * *Can the authors comment on the statistical significance of the results in Tables 2 and 3?*
>
> We acknowledge the need for clarity about statistical significance. The uncertainty values presented in Tables 2 & 3 represent 95-percent confidence intervals.
>
>
> Consider the cells highlighted in blue. HINTs-FC exhibits large intervals on the InvertedDoublePendulum task which can be visualised in Fig 3. All other tasks show moderate intervals that do not overlap with baselines.
>
>
> __On significance__ The performance results of HINTs-FC with good h (highlighted in blue) and PPO-RGB agents are statistically significant in all tasks. Moreover, the performance improvement over HINTs-FC-x (state-only hints) is statistically significant for HINTs-FC with human-identified hints.

---

### Official Review · Reviewer_Qu2o · 2025-11-01

**Soundness:** 2
**Presentation:** 2
**Contribution:** 2
**Rating:** 2
**Confidence:** 2

**Summary:**

The paper proposes HINTS, a framework that integrates human-intuited conceptual cues into reinforcement-learning (RL) training for continuous-control tasks under partial observability. Instead of demonstrations or full human supervision, the approach introduces a programmatic generator that emits grounded hints (e.g., curvature, angular velocity, or goal distance) representing the kind of coaching guidance a human might give. These hints condition the policy through four possible mechanisms—latent concatenation (LC), additive (AC), feature-wise affine (FC), and masked conditioning (MC). The authors empirically evaluate HINTS on image-based Classic Control, Car Racing, and MuJoCo Locomotion domains, demonstrating high sample efficiency and strong performance.

**Strengths:**

Compelling idea: Human‑intuitive coaching as grounded cues, bridging between pure RL and demo‑heavy IL.

Breadth of evaluation: Classic Control, CarRacing (including Hairpin distribution shift), and MuJoCo locomotion

Informative ablations: Utility of targeted hints over full/partial state, suggesting careful hint design matters.

Consistent gains under data limits: Notably in Classic Control and Humanoid, where HINTS trails only DAGGER but beats PPO‑RGB.

**Weaknesses:**

In the introduction, the contribution is motivated while comparing it to behaviour cloning methods. However, there are other human feedback methods in reinforcement learning. I encourage authors to position themselves using the survey paper here and make the necessary comparisons: https://www.frontiersin.org/journals/robotics-and-ai/articles/10.3389/frobt.2021.584075/full

Opaque generator design. The paper does not detail how G computes curvature, on-track flags, etc.(key for reproducibility).

No clear selection rule for conditioning schemes. LC/AC/FC/MC are task-dependent but unexplained; a unified comparison in the main text is missing. I would pick only FC in the main text, and explain alternatives with comparison in the appendix.

Missing code release

**Questions:**

How does HINTS differ from frameworks in feedback RL literature?

How are hints are generated during training?

Please clarify generator's design, does it access privileged state?

Can hints learned in one domain (e.g., curvature in Car Racing) transfer to related environments without retraining the generator?

How should one pick among LC/AC/FC/MC?

---

> ### Author Response · Authors · 2025-11-24
>
> Thank you for supporting our work and offering concrete suggestions to improve the clarity of our work. Please find our responses below:
>
>
> __positioning of our work__
>
>
> > How does HINTS differ from frameworks in feedback RL literature?
>
>
> Thank you for this question. Please review the extended related work in [our response to @o8SK](https://openreview.net/forum?id=W0tpFxWcdd&noteId=SC4hdxabv2) and (for the second point) additional comments in [our response in the official comments](https://openreview.net/forum?id=W0tpFxWcdd&noteId=qXuuyp29DZ).
>
>
> __implementation__
>
>
> > How should one pick among LC/AC/FC/MC?
>
>
> Our experiments showed consistent performance from learning with FC. We recommend selecting this  conditioning method for tasks. The analysis of other conditioning methods was useful for validating our observations about how performance depends on the conditioning information. We saw a consistent relationship across the different methods (Table 4).
>
>
> > Can hints learned in one domain (e.g., curvature in Car Racing) transfer to related environments without retraining the generator?
>
>
> Note that some tasks in our experiment suite share common structure. For example, the Acrobot and Pendulum tasks are solvable by "swinging" motions. In our framework, the human can coach the agent by relaying one concept to the agent who can then index into its learnt representation to quickly solve these tasks. The generator would ground this singular concept in different, but related ways for each environment (e.g., angular velocities). Since the cues generated are specific to the environment, the generator in our current implementation should be retrained. However, training the generator with supervision takes less than 10 minutes.
>
>
> > Missing code release
>
>
> We have published our anonymised codebase at https://github.com/charlie-the-brave/anon-hints.git.

---

### Official Review · Reviewer_6fts · 2025-11-04

**Soundness:** 2
**Presentation:** 3
**Contribution:** 2
**Rating:** 2
**Confidence:** 4

**Summary:**

This paper investigates the use of "human coaching" to enhance the learning efficiency of reinforcement learning agents. The authors introduce a hint generator that processes conceptual human hints and a hint-conditioned policy that integrates these hints. The proposed approach is evaluated on a selection of MuJoCo and Car Racing tasks. Experimental results demonstrate that the method outperforms baselines that do not incorporate human hints.

**Strengths:**

1. The motivation of this paper is clearly established. The capacity for a RL agent to integrate and utilize external conceptual hints is a functionally important yet empirically underdeveloped research domain. This capability is significant because it allows the learning process to move beyond simple reward maximization based solely on raw environmental feedback.

2. While some clarification is needed, this paper is generally well structured.

**Weaknesses:**

1. The reviewer has major concerns regarding the core technical contribution of this work. While the authors characterize the proposed methodology as Human-in-the-Loop Reinforcement Learning, the implemented mechanism appears to function primarily as advanced input feature engineering. The approach involves providing the agent with additional, privileged information as extended input features to the observation space, which facilitates the learning process. Critically, the reviewer notes a lack of genuine, dynamic interaction or feedback between the learning policy and the human operator, which is typically the defining characteristic of a true HiL system.

2. The proposed approach raises serious concerns regarding scalability and generalizability. A significant limitation is the reliance on manual intervention; specifically, for each new task, users are required to design and provide a distinct, tailored set of 'cues' or hints.

**Questions:**

The current experimental validation is confined to a restricted set of environments. To conclusively demonstrate the robustness and broader applicability of the proposed method, the experimental section requires significant expansion. Specifically, the inclusion of results from more diverse and complex environments—such as an expanded suite of Mujoco tasks, challenges from the DeepMind Control Suite (DM Control), or large-scale strategic domains like StarCraft II—would substantially strengthen the credibility of the findings.

---

> ### Author Response · Authors · 2025-11-24
>
> Thank you for agreeing with the motivation and presentation of our work. We value your critical feedback about the positioning of our approach. Please find our responses to your comments below:
>
> __positioning of our work__
>
> > 1. ... Critically, the reviewer notes a lack of genuine, dynamic interaction or feedback between the learning policy and the human operator.
>
> > 2. The proposed approach raises serious concerns regarding scalability and generalizability... For each new task, users are required to design and provide a distinct, tailored set of 'cues' or hints.
>
> We appreciate your request to clarify these two points. Our related work should have included a section on HIL learning to clarify what our approach contributes to this body of work. @Qu2o shared similar concerns with regard to feedback RL, which we consider a subarea of HIL. Please review the extended related work in [our response to @o8SK](https://openreview.net/forum?id=W0tpFxWcdd&noteId=SC4hdxabv2) and (for the second point) additional comments in [our response in the official comments](https://openreview.net/forum?id=W0tpFxWcdd&noteId=qXuuyp29DZ).
>
> To summarise, our approach integrates human input using a generator that grounds this information as cues, allowing the human to scaffold learning with structure shared across related tasks. Other frameworks focus on learning by imitation or reward inference which present expensive data collection challenges. Our contribution offers a good solution to scenarios in which data-scarcity is a primary factor.
>
> __extended experiments__
>
> > The current experimental validation is confined to a restricted set of environments.
>
> We agree with extending our experiments to more complex domains. DeepMind Control Suite is an interesting suggestion that we are exploring with alternative backbones like DrQV2 [[Yarats+21]](https://openreview.net/pdf?id=GY6-6sTvGaf). The training algorithm for the generator will need to be upgraded to handle challenges in the grounding problem like multimodality, self-supervision, etc.
>
> Yarats et al. Image Augmentation Is All You Need: Regularizing Deep Reinforcement Learning from Pixels. ICLR 2021.

---

### Author Response · Authors · 2025-11-19
**Thank you reviewers - an initial response**

### overview


Thank you to the reviewers for taking the time to give constructive feedback on our work. We feel encouraged by your informative suggestions and questions to continue iterating on this work. We are excited to begin a productive discussion with you.


Our initial response addresses comments about the presentation and a key point raised by @rFfh and @6fts about our contribution. We will follow up with responses to questions by @Qu2o and @o8SK about implementation details and related work.

__Please refer to https://github.com/charlie-the-brave/anon-hints/blob/reviews/reformat/revisions.pdf for details__.

---

__formatting__


We have reformatted the tables and figures pointed out by @o8SK. We will continue iterating based on further feedback. As a start, we have updated the table captions and applied colors to table cells to improve readability. Figure 3 has been reformatted. Please see the following samples:

Table 2 with more neatly spaced cells, colors to distinguish cells, and updated caption. [see PDF Tab. 1]


We restructured Figure 3 and enlarged the text. [see PDF Fig. 2]

---

__questions__


> *Is the hint just a handcrafted feature? [[@rFfh](https://openreview.net/forum?id=W0tpFxWcdd&noteId=2uUkR4A8fN),[@6fts](https://openreview.net/forum?id=W0tpFxWcdd&noteId=OtDDj61PHI)]*


Thank you for raising this question as it is critical to framing our work. Hints are not handcrafted features, they are signals that scaffold the agent's own learning of representations, value estimates, and policies.


A. One possible definition of hand-crafted features could be state features crafted by a human.


Tables 2 & 3 show examples of action- and reward-based grounded hints (Pendulum and Cheetah). As an example, the human can specify a conceptual hint to "swing-up". This concept can be grounded in multiple ways in the state or action space through state- or action-based cues shown along the rows in Table 2.


B. Considering a more general definition, hand-crafted features could be manually-defined state representations, goal directives, reward shaping schemes, exploration heuristics, etc. Hand-crafted features often replace components that could otherwise be learnt; e.g., via feature learning, state representation, value estimation.


With HINTs, an expert human does more than provide hand-crafted features. The human in our framework can ground structural information via the generator G which reveals useful learning signals. What we uniquely show with HINTs is the ability for the human to identify structures that help the agent index task relevant information from its observations. Via our framework, humans can guide learning more scalably than existing imitation learning and IRL frameworks which either require a large number of demonstrations ([Hejna+23](https://arxiv.org/abs/2306.12554)) or high quality demonstrations ([Szot+23](https://arxiv.org/abs/2303.16194)).


Concretely, if we train agents on the hints alone, they do not perform as well as agents trained with our framework (see Fig 4a: car racing blind agent and Fig R1: Pendulum and InvertedDoublePendulum with blind agents). This suggests that it is necessary for the agent to learn its own representation as there is not enough information in the hint alone.

Fig 4a: performance of car racing blind agents versus PPO-RGB and HINTs agents


[seen here](https://github.com/charlie-the-brave/anon-hints/blob/master/reformat/Figure4a_car_blind.png?raw=true)


Figure R1: eval performance of PPO-x (state-only; green) and PPO-h (cue-only; orange) versus HINTs agents (blue)


Penulum [seen here](https://github.com/charlie-the-brave/anon-hints/blob/master/reformat/Pend_rewards_angle_vector_train.png?raw=true)


InvertedDoublePendulum [seen here](https://github.com/charlie-the-brave/anon-hints/blob/master/reformat/IDP_rewards_composite_vector_train.png?raw=true)




## references


*Hejna et al. Improving Long-Horizon Imitation Through Instruction Prediction. AAAI 2023.*


*Szot et al. BC-IRL: Learning Generalizable Reward Functions from Demonstrations. ICLR 2023.*

---

> ### Author Response · Authors · 2025-11-21
> **Follow up about implementation details and related work**
>
> We appreciate each reviewer suggesting clarifications on the methods (Section 3) and related work (Section 7). We will share our revisions in a linked pdf.
>
> __implementation details__ [@Qu2o,@rFfh,@o8SK]
>
> Many reviewers pointed out the need for clarification on the low-level implementation of the generator. __We have released the code__ at https://github.com/charlie-the-brave/anon-hints/tree/reviews and a pdf containing implementation details at https://github.com/charlie-the-brave/anon-hints/blob/reviews/reformat/revisions.pdf. These will be provided as part of the appendix in the next iteration.
>
> __positioning of our work__
>
> > How does HINTS differ from frameworks in feedback RL literature? [[@Qu2o](https://openreview.net/forum?id=W0tpFxWcdd&noteId=ZUKVRjBC3V)]
>
> > In the introduction, the contribution is motivated while comparing it to behaviour cloning methods. However, there are other human feedback methods in reinforcement learning [[Qu2o](https://openreview.net/forum?id=W0tpFxWcdd&noteId=ZUKVRjBC3V)]
>
> > Critically, the reviewer notes a lack of genuine, dynamic interaction or feedback between the learning policy and the human operator, which is typically the defining characteristic of a true HiL system. [[6fts](https://openreview.net/forum?id=W0tpFxWcdd&noteId=OtDDj61PHI)]
>
> Please read [our response to @o8SK](https://openreview.net/forum?id=W0tpFxWcdd&noteId=SC4hdxabv2) who raised a similar question.
>
> To highlight the advantage of our method over iterative feedback methods, consider the following:
> Note that some tasks in our experiment suite share common structure. For example, the Acrobot and Pendulum tasks are solvable by "swinging" motions. In our framework, the human can coach the agent by relaying one concept to the agent who can then index into its learnt representation to quickly solve these tasks. The generator would ground this singular concept in different, but related ways for each environment (e.g., angular velocities). With iterative feedback mechanisms, the human would waste effort providing preference labels, demonstrations, or corrections, etc. for each task.

---

### Meta-Review · Area_Chair_axDd · 2026-01-05

**Summary:**

HINTS proposes putting humans in the RL loop as coaches. Experts provide conceptual tips that are turned into programmatically generated, human intuited cues, conditioning the policy without prescribing full solutions or requiring clean demonstrations. Multiple hint conditioning schemes are described to cover control tasks with varying visual complexity, targeting data constrained visual continuous control under partial observability. Across classic control, car navigation and goal reaching, & locomotion, hint conditioned agents are reported to learn faster than vision only baselines and to transfer better to harder variations, with composite and concise hints often strongest. Overall, the approach aims to yield more broadly applicable and reliable stratrgies under tight training budgets, sometimes narrowing the gap to state based agents trained with large scale experience or costly supervision.

In the original round of reviews the committee acknowledged the motivation and practical appeal of injecting human intuited, grounded cues into pixel based RL and the reported sample efficiency gains & ablations (6fts, qu2o, rffh, o8sk). The committee also raised consistent concerns that the core technical contribution may largely reduce to task specific, privileged feature engineering via a hardcoded or opaque hint generator rather than human in the loop interaction or feedback (reviewer 6fts, rffh, qu2o). Scalability and generalizability were questioned since new tasks appear to require manual design of tailored cues, and transfer of hints or the generator is not clearly established (reviewers: 6fts, qu2o, o8sk). Clarity and reproducibility were also flagged, including missing low level generator details, uncertainty about privileged state access, lack of code at review time, and presentation and results reporting issues such as unclear conditioning scheme choice, confusing tables and figure placement, and questions about statistical significance (reviewers qu2o, rffh, o8sk).

The rebuttal includes an initial response and detailed follow ups, along with anonymized links to reformatted figures and tables and an anonymized code release. The area chair concurs with tthe consensus of the four original reviews that these updates primarily improve clarity, while leaving the main points of evaluation only partially addressed. In particular, the current evidence still suggests reliance on environment and task specific cue design and substantial use of state derived groundings in experiments, so scalability, transfer, and claims around avoiding privileged information remain insufficiently supported within the scope of the submission. The expanded related work and the suggested default conditioning choice help with positioning, but a principled conditioning selection rule and a clean separation from feature injection baselines remain under justified, and broader validation is largely deferred rather than demonstrated here.

**Reviewer Concerns:**

The rebuttal and follow-up comments primarily strengthened clarity and reproducibility: additional implementation pointers were provided, along with an anonymized code link and a reformatted manuscript that improves the presentation of figures/tables and expands related work (touching several of the presentation and missing-detail points raised across reviews). Several of the central concerns from the initial reviews, however, remain only partially resolved within the scope of the submission and evidence: the current instantiation still appears to depend on task- and environment-specific cue design and on how those cues are obtained during training/evaluation; the extent to which the approach cleanly separates from 'adding extra signals' baselines, or supports the broader framing around human-in-the-loop feedback and generalization, is not fully demonstrated; and broader validation beyond the current suite is largely discussed as future work rather than shown.

**Reviewer Scores:**

Since this ICLR's discussion period was cut short (Nov 28), a score-change estimate is inherently uncertain.
Despite this: based on the original review justifications and the rebuttal’s focus on clarifications, it seems more likely that scores (originally all at 2) would have stayed similar or shifted only modestly (mostly on clarity/presentation), without moving to a clear acceptance level (6 or higher). By-reviewer: 6fts likely unchanged; qu2o, rffh, and o8sk possibly unchanged or slightly higher, but not enough to change the overall consensus/outcome.

---

### Decision · Program_Chairs · 2026-01-26

Reject